



# Snowmelt Characterization from Optical and Synthetic Aperture Radar Observations in the Lajoie Basin, British Columbia

Sara E. Darychuk[1], Joseph M. Shea[1], Brian Menounos[1,2], Anna Chesnokova[1], Georg Jost[3] & Frank Weber[3]

[1]Deptartment of Geography, Earth and Environmental Sciences, University of Northern British Columbia, Prince George, V2N 4Z9, Canada
[2]Hakai Institute, Campbell River, British Columbia, Canada
[3]BC Hydro, Burnaby, British Columbia, Canada V3N 4X8

Correspondence to: S. Darychuk (darychuk@unbc.ca)

**Abstract.** Snowmelt runoff serves both human needs and ecosystem services and is an important parameter in operational forecasting systems. Sentinel-1 Synthetic Aperture Radar (SAR) observations can estimate the timing of melt within a snowpack; however, these estimates have not been applied on large spatial scales. We present here a workflow to fuse Sentinel-1 SAR and optical data from Landsat-8 and Sentinel-2 to estimate the onset and duration of snowmelt in the Lajoie Basin, a large watershed in the Southern Coast Mountains of British Columbia. A backscatter threshold is used to infer the point at which snowpack saturation occurs, and the snowpack begins to produce runoff. Multispectral imagery is used to estimate snow free dates across the basin to define the end of the snowmelt period. SAR estimates of snowmelt onset form consistent trends by elevation and aspect on the watershed scale and reflect snowmelt records from continuous SWE observations. SAR estimates of snowpack saturation are most effective on moderate to low slopes ($< 30°$) in open areas. The accuracy of snowmelt durations is reduced due to persistent cloud cover in optical imagery. Despite these challenges, snowmelt durations agree with trends in snow depths observed in the Lajoie. This approach has high potential for adaptability to other alpine regions and can provide estimates of snowmelt timing in ungauged basins.

## 1 Introduction

Snowmelt runoff is an important source of streamflow or groundwater recharge in many regions of the world. Snowmelt comprises approximately 32% of global freshwater discharge (Meybeck et al., 2001), and over one billion people depend on this seasonal water source (Barnett et al., 2005). Both the volume and persistence of the snowpack are altered by warming global temperatures (Mote et al., 2005). Changes to snowmelt timing will broadly impact ecosystem health and natural hazard frequency, as the timing of snowmelt is linked with trends in soil moisture (Harpold et al., 2015; Kampf et al., 2015), streamflow (Luce and Holden, 2009; Rauscher et al., 2008; Déry et al., 2009), and wildfire (Westerling, 2016; Westerling et al., 2006). To predict the release of water from a snowpack, a thorough understanding of snowmelt is required.



Snowmelt occurs during three phases: moistening, ripening, and runoff (Dingman, 2015). Snowmelt begins with moistening, when the top layers of the snowpack start to melt due to increases in air temperatures or solar radiation. When the wetting front penetrates the snowpack, the ripening phase begins. Once the snowpack is fully saturated and isothermal (0°C throughout), further energy inputs will be directed at snowmelt and liquid water will be released (Dingman, 2015). The phases of snowmelt can be identified through continuous measurements of liquid water content (LWC) and snow water equivalence (SWE). The

runoff phase initiates when a snowpack has reached its maximum liquid water content and SWE sharply declines.

Several methods exist to monitor SWE and snowpack LWC. SWE is measured manually via snow sampling tubes or automatically via snow scales and snow pillows (Kinar and Pomeroy 2015). Manual and automated measurements of LWC use variable techniques, including dielectric, centrifugal, or calorimetric methods (Kinar and Pomeroy 2015). Centrifugal and calorimetric methods are time consuming and destructive, whereas dielectric techniques are more common for in-situ

observations of LWC. Systems that exploit dielectric properties in the microwave region of the electromagnetic spectrum, such as the snowpack analyzer (Stähli et al., 2004) or upward-looking ground-penetrating radar (Schmid et al., 2014), can provide automated LWC measurements for operational forecasting. Establishing networks of continuous SWE and LWC observations is labour intensive and expensive in high alpine areas, which are difficult to access and can experience extreme winter conditions. Due to these time and cost constraints, physical snowpack observations are often limited to point observations at

mid-elevations of mountainous ranges, and are insufficient to capture the spatial variability of the snowpack.

Snow cover is highly variable from the watershed to sub-grid scale (Deems et al., 2006; Lopez-Moreno et al., 2015), and point observations do not always represent snowpack conditions in nearby areas (Elder et al., 1991; Neumann et al. 2006). Further, high elevation snowpack observations are crucial, as headwater regions generate large proportions of baseflow in mountain streams (Rumsey et al., 2020). In the province of British Columbia, Canada, there are 120 automated snow weather stations

that monitor 22 major watersheds (Government of British Columbia, 2017). These stations are situated at an average elevation of 1400 m above sea level (asl), and less than 10% are located above 2000 m asl to monitor high alpine snowpacks (i.e., the Coast and Rocky Mountains of British Columbia reach elevations of 4000 m asl and 3300 m asl, respectively). The low density of physical observations in montane regions, such as those in British Columbia, hinder snowpack monitoring and modelling efforts. Distributed observations of the snowpack are required for accurate snowmelt runoff forecasting and model

improvement (Luce et al., 1998). Considering the need for spatially distributed observations of the snowpack, remote sensing observations are attractive supplements to physical monitoring systems.

Remote sensing data offer alternative methods to monitor snowmelt and estimate the timing of runoff from the snowpack. Marin et al. (2020) demonstrated the sensitivity of Sentinel-1 SAR to SWE and snowpack LWC at five test sites in the European Alps. Three phases of melting were identified from the SAR time series at point locations. The onset of snowmelt

runoff, or the point in time when LWC is at its maximum and SWE sharply declines, coincided with minima in SAR time series at all test locations. SAR is effective for monitoring snowmelt due to its sensitivity to the LWC of snow; however, SAR signals will be broadly impacted by landcover type and topography.





C-band SAR signals (such as those from Sentinel-1) can penetrate dry snowpacks to depths of several meters (Mätzler, 1987). In a dry snowpack, the backscatter signal is the sum of volume scattering within the snowpack and surface scattering at the snow/ground interface, however, surface scattering is the dominant signal (Shi and Dozier, 1995; Nagler and Rott, 2000). In a wet snowpack, liquid water becomes the dominant factor influencing backscatter and surface scattering can be neglected (Shi and Dozier, 1995; Nagler and Rott, 2000). As demonstrated by Marin et al. (2020) backscatter values become increasingly negative as LWC increases during the ablation season, reaching a minimum at the point of runoff (or melt) onset. After reaching their minima, backscatter values increase until the snowpack dissipates (Marin et al., 2020). However, the relation between SAR and snowmelt will be impacted by physical properties of the snowpack and local terrain. SAR signals are altered by snowpack properties such as snow grain size, density, depth, stratigraphy, impurity content, and surface roughness (Liu et al., 2006). Further, the magnitude of SAR backscatter values over snow-covered regions will be impacted by local incidence angle (Nagler and Rott, 2000; Shi and Dozier, 1995), forest cover (Pivot, 2012), and polarization (Nagler et al., 2016). The influence of snowpack characteristics and topography on SAR estimates of snowmelt onset requires further exploration.

In British Columbia, where mountain ranges and winter snowpacks dominate the landscape, remote sensing observations are needed to characterize snowmelt as few physical snowpack measurements are available. In this paper, we fuse C-band Sentinel-1 SAR imagery with Landsat-8 and Sentinel-2 multispectral imagery to define the onset and duration of snowmelt in the Lajoie Basin, a large watershed in the Southern Coast Mountains of British Columbia. The relation between SAR minima and snowmelt established by Marin et al. (2020) is examined over varied polarizations, land cover classes, aspects, and hillslopes. Estimates of snowmelt onset and duration are verified with continuous records of SWE. This work establishes a low-cost and adaptable method for monitoring snowmelt onset in ungauged basins that can be used by watershed managers across western North America.

## 2 Study Area

The Lajoie Basin is a 985 km$^2$ watershed located in the Coast Mountains of British Columbia, 170 km north of Vancouver. Runoff in the watershed flows into Downton Lake, which is formed by the Lajoie Dam. The Lajoie is a critical area for hydroelectric power generation as the Lajoie Dam is the first structure in a larger hydroelectric system that generates approximately 5% of the province's power. The watershed ranges in elevation from 800 m asl to 2800 m asl and has a median elevation of 1910 m asl. The lower elevations of the Lajoie are forested, and the treed area covers 47% of the catchment (Figure 1). The dominant biogeoclimatic zones in the watershed are Engelmann Spruce – Subalpine Fir and Montane Spruce, with small areas of Interior Mountain Heather Alpine and Boreal Altai Fescue Alpine zones (Pojar et al., 1987). Forest harvesting occurs within the watershed to a minor extent, with less than 10% of the basin impacted.

Precipitation records from the Green Mountain automated weather station (Figure 1) indicate this region received an average of 1090 mm of precipitation between 1993 and 2011. During this period, approximately 39% of precipitation fell as snow and



SWE values exceeded 1300 mm. Of the total basin area 16% is glacierized (Figure 1). As a result, the basin has a nival-glacial
hydrological regime and is mostly fed by snowmelt in May and June and glacier melt in July and August.

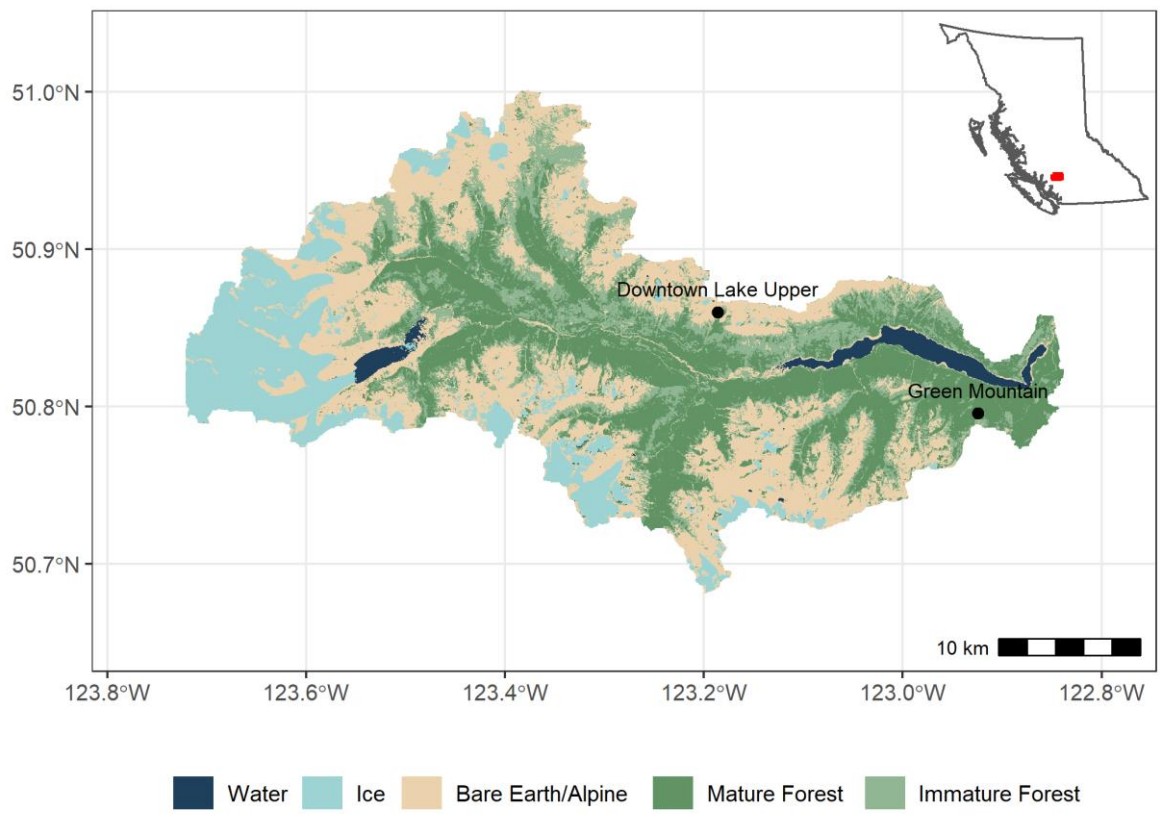

**Figure 1.** The Lajoie Basin by land cover type. Land cover types were determined via a Random Forest classification in Google Earth
Engine from Sentinel-2 imagery.

## 3 Data

**3.1 Telemetry & LiDAR Data**

Two automated weather stations located in the Lajoie Basin are operated by the provincial hydroelectric utility BC Hydro
(Figure 1). The Downton Lake Upper automated weather station (1829 m asl) and Green Mountain automated weather station
(1725 m asl) provide hourly records of temperature, precipitation, and SWE. Downtown Lake Upper is situated on a northwest
facing slope, and is classified as disturbed forest (Figure 1). Green Mountain is situated on a northeast facing slope and is
classified as mature forest (Figure 1).





A 5 m bare-earth LiDAR DEM was used to create slope, aspect, and elevation classes in the Lajoie Basin. LiDAR data was collected September 22nd, 2017 from a fixed wing aircraft. The DEM was resampled to 30 m and classified into four aspect classes, six slope classes, and twenty elevation classes.

### 3.2 Sentinel-1 SAR

The Sentinel-1 constellation acquires SAR scenes every two to three days over the Lajoie Basin. The constellation consists of two satellites, each orbiting the Earth in opposing directions (ascending or descending). Each satellite has two tracks that capture the Lajoie. In the descending direction both tracks (13 and 86) completely capture the Lajoie; however, in the ascending direction only one track (137) has full coverage while the other (64) captures the east side of the Basin only (Table 1). Local incidence angles vary slightly by satellite track (Table 1). On average, incidence angles are higher for tracks in the descending

direction (Table 1). Local incidence angles in descending Sentinel-1 images at Downtown Lake Upper measure between 42° and 33°, whereas at Green Mountain they measure between 34° and 26°.

**Table 1.** Sentinel-1 SAR orbit tracks over the Lajoie Basin. Percentiles are for average Local Incidence Angles for each track are displayed, as well as the area covered by that track (calculated as a percent of the total basin area).

| Direction | Track | 5th Percentile | 50th Percentile | 95th Percentile | % Covered |
|---|---|---|---|---|---|
| Ascending | 64 | 11.2° | 35.2° | 61.8° | 53 |
| Ascending | 137 | 18.8° | 42.8° | 68.2° | 100 |
| Descending | 13 | 22.2° | 44.3° | 71.7° | 100 |
| Descending | 86 | 15.7° | 36.8° | 63.7° | 100 |


Images over the watershed are acquired in the Interferometric Wide Swath mode, the default collection mode over land. These SAR scenes have a resolution of 10m and have two polarization bands available, VV (Vertical-Vertical) and VH (Vertical-Horizontal).

The Level-1 GRD sentinel-1 SAR dataset from GEE is pre-processed using the Sentinel-1 Toolbox (European Space Agency,

2021). The images undergo five steps during correction:

1. Apply precise orbital file
2. Remove ground range detected (GRD) border noise
3. Remove thermal noise
4. Convert digital numbers to backscatter ($\sigma°$) in decibels (dB)
5. Correct for terrain (Range-Doppler method)

The Committee on Earth Observation Society Analysis-Ready-Data for Land identifies additional steps needed for SAR pre-processing (Lewis et al., 2018) that are not included in the Senintel-1 GEE catalogue. These steps include radiometric terrain correction and pixel-based identification of shadow or layover (Lewis et al., 2018).

## 3.3 Multispectral & Optical Data

The Operational Land Imager and Thermal Infrared Sensor aboard Landsat 8 from the National Aeronautics and Space Administration acquire imagery over the Lajoie Basin every 16 days. Images are provided at a 30 m resolution and bands provide coverage across the visible, near infrared, shortwave infrared and thermal spectrums. Top-of atmosphere (TOA) reflectance Landsat 8 scenes are publicly available in the GEE data catalogue.

The MultiSpectral Instrument aboard Sentinel-2 from the European Space Agency collects multispectral imagery over the Lajoie Basin every six days. Images are provided at 20 m resolution and bands provide coverage across the visible, red-edge, near infrared, shortwave infrared and thermal spectrums. TOA reflectance Sentinel-2 scenes are freely available in the GEE data catalogue.

## 4 Methods

### 4.1 Snowpack Saturation from Sentinel-1 SAR

Dates of snowpack saturation from 2018 to 2021 were estimated from Sentinel-1 SAR images. GEE was used to access, correct, and download Sentinel-1 SAR scenes. To detect the onset of snowmelt in the basin, we restricted our analysis for SAR scenes acquired between February and August. We created six image collections from Sentinel-1 SAR scenes, with two collections created for each study year based on the directional pass of the satellite (ascending or descending) (Table 2). To prevent border noise from impacting results, we only selected images containing the whole basin (i.e., those captured by tracks 13, 86, and 137) to create the stacks. We corrected the Sentinel-1 image collections for slope-induced radiometric distortion using a volumetric model (Vollrath et al., 2020), and applied a filter to reduce radar induced speckle (Lopes et al., 1990). After corrections, we resampled the image collections to 30 m for consistent analysis with optical imagery and downloaded. We performed these methods for images captured in VV and VH polarization.



**Table 2.** The number of Sentinel-1 SAR images used for analysis by satellite pass direction.

| Year | Pass Direction | Number of Images | Image Frequency |
|------|----------------|------------------|-----------------|
| **2018** | Ascending | 24 | 12-36 days |
| **2018** | Descending | 38 | 5-7 days |
| **2019** | Ascending | 40 | 12 days |
| **2019** | Descending | 39 | 5-7 days |
| **2020** | Ascending | 40 | 12 days |
| **2020** | Descending | 40 | 5-7 days |
| **2021** | Ascending | 40 | 6-12 days |
| **2021** | Descending | 44 | 1-7 days |

We used RStudio, an integrated development environment for R (RStudio Team, 2021), to estimate snowpack saturation from
the created Sentinel-1 image collections. Before we extracted saturation dates, we temporally smoothed time series of
backscatter values in each pixel using a Locally Weighted Least Squares Regression with a span of 0.2 (Supplementary
Materials). This correction reduces noise in SAR time series, and thus minimizes false positives in the final saturation date
maps. After smoothing, we identified the date of the minimum backscatter value for each pixel.

We used backscatter minima to approximate the date of snowpack saturation (Marin et al., 2020) for all pixels in the basin,
regardless of their snow-covered status. After we extracted saturation dates, we calculated mean values for each elevation,
slope, aspect, and land cover class. To correct for any remaining artifacts of SAR speckle and distortion, we removed any
pixels with estimated saturation dates that were outside of two and a half standard deviations of the calculated means. We then
infilled saturation dates with the median value from that pixel's elevation, slope, aspect, and land cover class (Supplementary
Materials).
To further validate the behaviour of SAR signals over complex terrain, we focus on SAR time series extracted from pixels
between elevations of 1600 m and 1800 m, and average them by land cover type and slope. We also extracted SAR time series
at the Green Mountain and Downton Lake Upper automated weather stations, where continuous measurements of SWE are
available. To create SAR signals at the location of each automated weather station, we used the average pixel values from all
pixels located within a 40 m circular radius of each station. All SAR time series were smoothed with a loess regression with a
span of 0.2.

**4.2 Snow Disappearance from Landsat-8 and Sentinel-2**

We determined snow disappearance from Sentinel-2 and Landsat-8 images in GEE. We filtered Sentinel-2 and Landsat-8
scenes were to create collections of images containing the Lajoie Basin captured between February and September from 2018
to 2021 (Table 3). We masked images for clouds and fused them into a single image collection for each study year (Table 3).




**Table 3.** The number of Sentinel-2 and Landsat-8 images used for analysis.

| Year | Landsat-8 Scenes | Sentinel-2 Scenes | Total |
|------|------------------|-------------------|-------|
| **2018** | 33 | 83 | 116 |
| **2019** | 36 | 79 | 115 |
| **2020** | 35 | 87 | 122 |
| **2021** | 28 | 95 | 123 |

We utilized a hybrid approach to approximate snow disappearance from the fused images. Over non-glacierized areas, we calculated Normalized Difference Snow Index (NDSI; Hall et al., 1995) and Normalized Difference Forest Snow Index

(NDFSI; Wang et al., 2015) values to classify each image for snow cover using a threshold of 0.4 (Table 4). For glacierized areas of the Lajoie Basin, we used a K-means clustering algorithm to classify the the initial image collection for snow cover. We created two classes on the glacier, snow and ice, based on reflectance values in the visible, near infrared, and shortwave infrared spectrums (Shea et al., 2013).

For all areas in the basin, we selected the first snow free date per pixel and extracted the corresponding date. We then exported

the maps of snow disappearance, or snowmelt end, from GEE and calculated mean values for each elevation, slope, aspect, and land cover class. To correct for errors to due clouds and cloud masking, we removed any pixels with estimated snow disappearance dates that were outside of two and a half standard deviations of the calculated means. We then infilled snow disappearance dates with the median value from that pixel's elevation, slope, aspect, and land cover class (Supplementary Materials).


**Table 4.** Vegetation indices used to detect snow cover from optical and multispectral imagery.

| Index | Formula | Source |
|-------|---------|--------|
| Normalized Difference Snow Index (NDSI) | $\dfrac{green - swir2}{green + swir2}$ | (Hall et al., 1995) |
| Normalized Difference Forest Snow Index (NDFSI) | $\dfrac{nir - swir2}{nir + swir2}$ | (Wang et al., 2015) |

### 4.3 Snowmelt Duration Estimates

We approximated snowmelt durations from the created maps of snowpack saturation and snowmelt end. We took the difference

in days between disappearance and saturation estimates as the duration of melt production for each study year. Elevation-based





thresholds were used to identify improbable melt durations, such as negative durations or durations exceeding 120 days. Improbable snowmelt durations were removed infilled with the median value from that pixel's elevation, slope, aspect, and land cover class (Supplementary Materials).

### 4.4 Telemetry Data

To validate SAR time series, estimates of snowmelt onset, and estimates of snowmelt duration we used continuous SWE records from the Downton Lake Upper and Green Mountain automated weather stations. We averaged hourly SWE measurements per day and calculated maximum SWE for each snow season between 2018 and 2021. We used a piecewise linear regression to infer the period of melt at each station. Regression models were determined using the *segmented* package (Muggeo, 2017), which implements a bootstrap restarting algorithm (Wood, 2001) to estimate breakpoints in the time series.

We used the first breakpoint after maximum SWE to approximate melt onset, and the last breakpoint to approximate the end of melt (Figure 2).



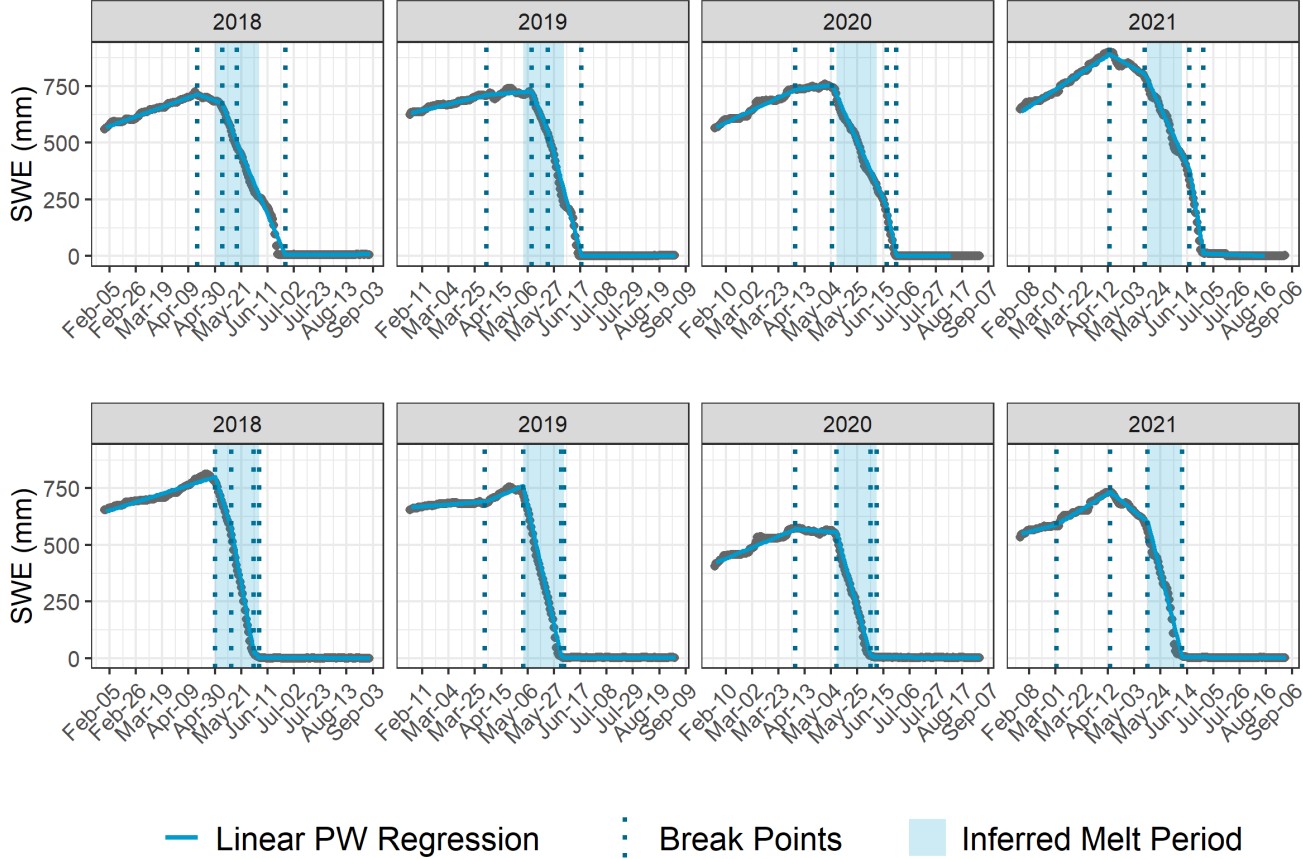

**Figure 2.** Snow Water Equivalence Records from Downton Lake Upper (Top) and Green Mountain (Bottom). Linear piecewise regressions of SWE trends are displayed with observational data. Breakpoints in the regression are marked with vertical dotted lines, and the inferred

melt period from the regression is shaded in light blue.

## 5 Results

### 5.1 Snowmelt from SWE Records

Estimates of snowmelt onset and duration from the linear piecewise regression were consistent during the study period. At Downton Lake Upper, snowmelt durations ranged from 40 to 51 days between study years with the shortest duration observed

in 2019. Snowmelt onset dates were consistent at Downton Lake Upper, ranging from May 4th to May 10th with the earliest onset recorded in 2018.  At Green Mountain, snowmelt durations ranged from 27 to 35 days between study years with the shortest duration estimated in 2021. Estimates of melt onset at Green Mountain ranged from April 29th to May 13th with the earliest onset observed in 2018.



## 5.2 Sensitivity of SAR to Snowmelt

At automated weather stations the minimum value in each SAR time series coincided with rapid (10-20 mm/day) declines in snow water equivalence (Figure 3, Figure 4). At Downton Lake Upper, SAR time series minima occurred within 0-10 days of SWE melt onset estimates for both polarizations (Figure 3). Comparing across four melt seasons at Downton Lake Upper, backscatter minima from VV polarized time series provided the least accurate approximations of melt in 2019 and the most accurate approximations of melt in 2018. VH polarized time series at Downton Lake Upper provided the least accurate

approximations in 2020 and the most accurate approximations in 2018. At Green Mountain, SAR minima occurred within 1-13 days of SWE melt onset estimates (Figure 4). Between study years, VV polarized time series at Green Mountain provided the least accurate approximations of melt in 2019 and the most accurate approximations in 2020. VH polarized time series at Green Mountain provided the least accurate approximations of melt in 2020 and the most accurate approximations in 2021. Minima from VV polarized images produced more accurate approximations of melt onset compared to those produced from

VH polarized images at both stations. At Downton Lake Upper, all VV-derived minima occurred within seven days of SWE melt onset estimates. At this location, only 50% of minima from VH polarized images occurred within seven days of melt onset. At Green Mountain, 75% of VV-derived minima occurred within seven days of SWE melt onset estimates. Similarly to Downton Lake Upper, only 50% of minima at Green Mountain from VH polarized images occurred within seven days of melt onset.






**Figure 3.** Comparisons of SAR time series and continuous snow water equivalence records (SWE) at Downtown Lake Upper. SWE records are displayed in dark blue, with melt periods shaded in light blue. SAR time series for VV and VH polarized images depicted in green, with time series minima denoted by dashed lines.





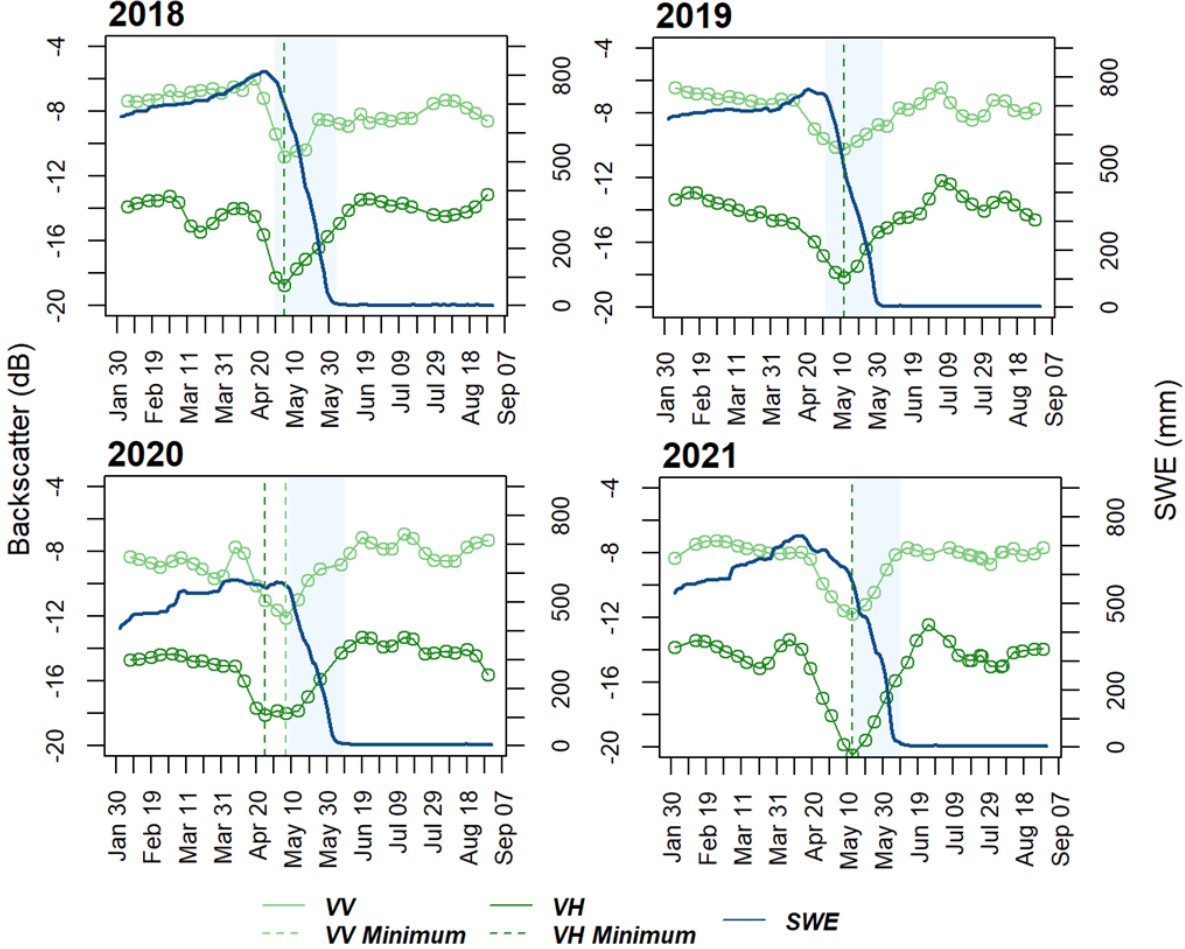

**Figure 4.** Comparisons of SAR time series and continuous snow water equivalence records (SWE) at Green Mountain. SWE records are displayed in dark blue, with melt periods shaded in light blue. SAR time series for VV and VH polarized images depicted in green, with time series minima denoted by dashed lines.

SAR backscatter time series from across the basin have a characteristic 'U' shape with their lowest values occurring in late spring (Figure 5). In open areas, the seasonal decrease is most pronounced on gentle slopes (< 10°) and decreases in amplitude as slope increases. Under mature forest cover SAR time series are reversed in terms of slope. In forested regions of the Lajoie, the seasonal decrease in backscatter showed the greatest amplitude on steep slopes (40-49°). Between land cover classes, SAR minima are the most pronounced over bare earth and become harder to detect under dense vegetation, especially on moderate (10 - 29°) slopes.

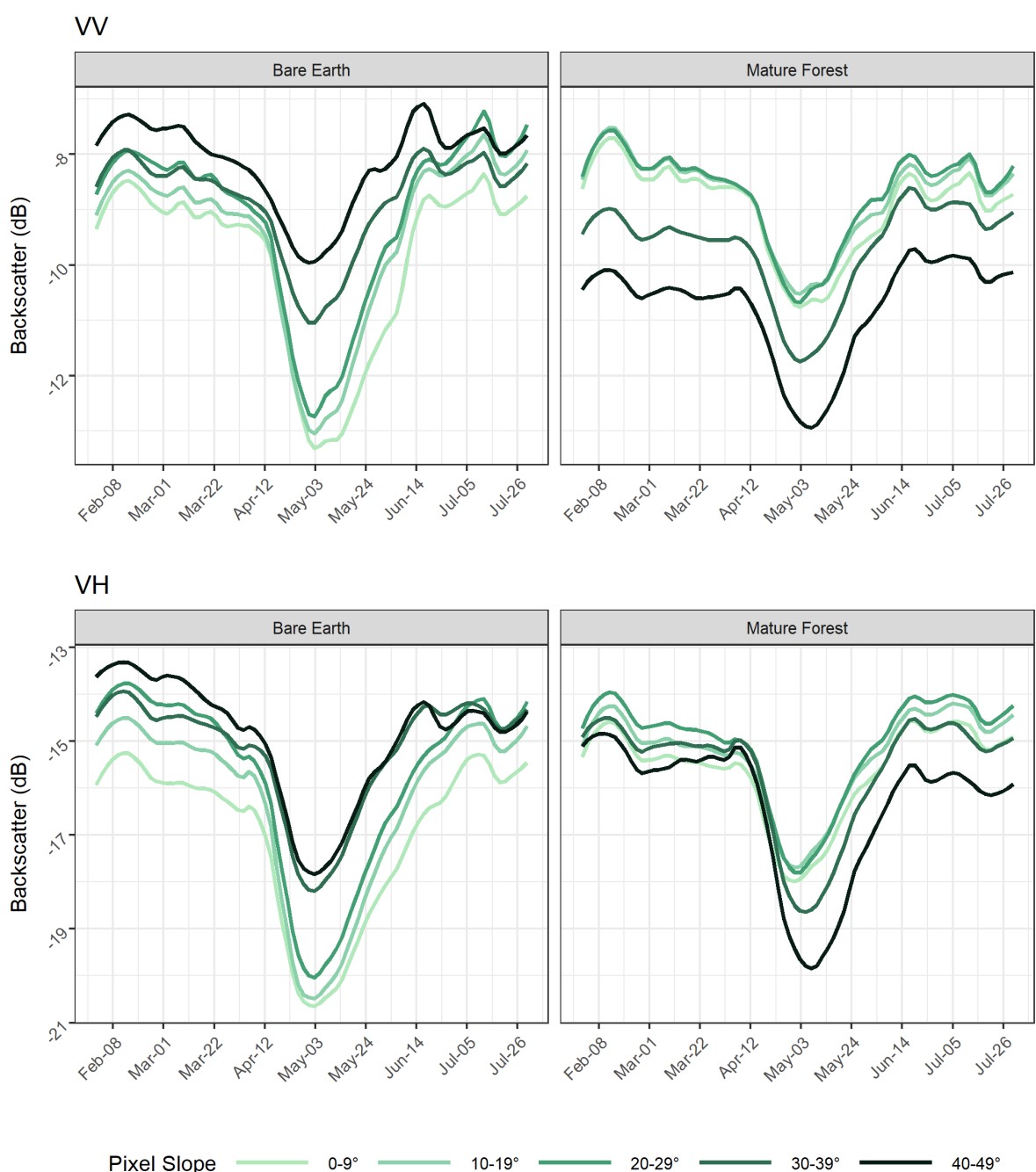


**Figure 5.** SAR backscatter time series in the Lajoie Basin from pixels located between 1600 and 1800 m from VV (top) and VH (bottom) polarized images. Observations under mature forest cover are displayed on the right, whereas observations in open areas are displayed on the left. Average backscatter for each cover type is shown by the shaded lines, with each line representing a different slope category. Observations are from 2021.



## 5.3 SAR Estimates of Melt Onset in the Lajoie

Due to the lower frequency of images in the ascending direction, results are shown for imagery captured by the descending pass of the satellite only. As time series from VH polarized images produced less reliable estimates of melt onset at Green Mountain and Downton Lake Upper, only results from VV polarized images are shown. For results from VH polarized images, see the Supplementary Materials. From 2018 to 2021 in the Lajoie, onset estimates were consistent by elevation with slight interannual variability (Figure 6).

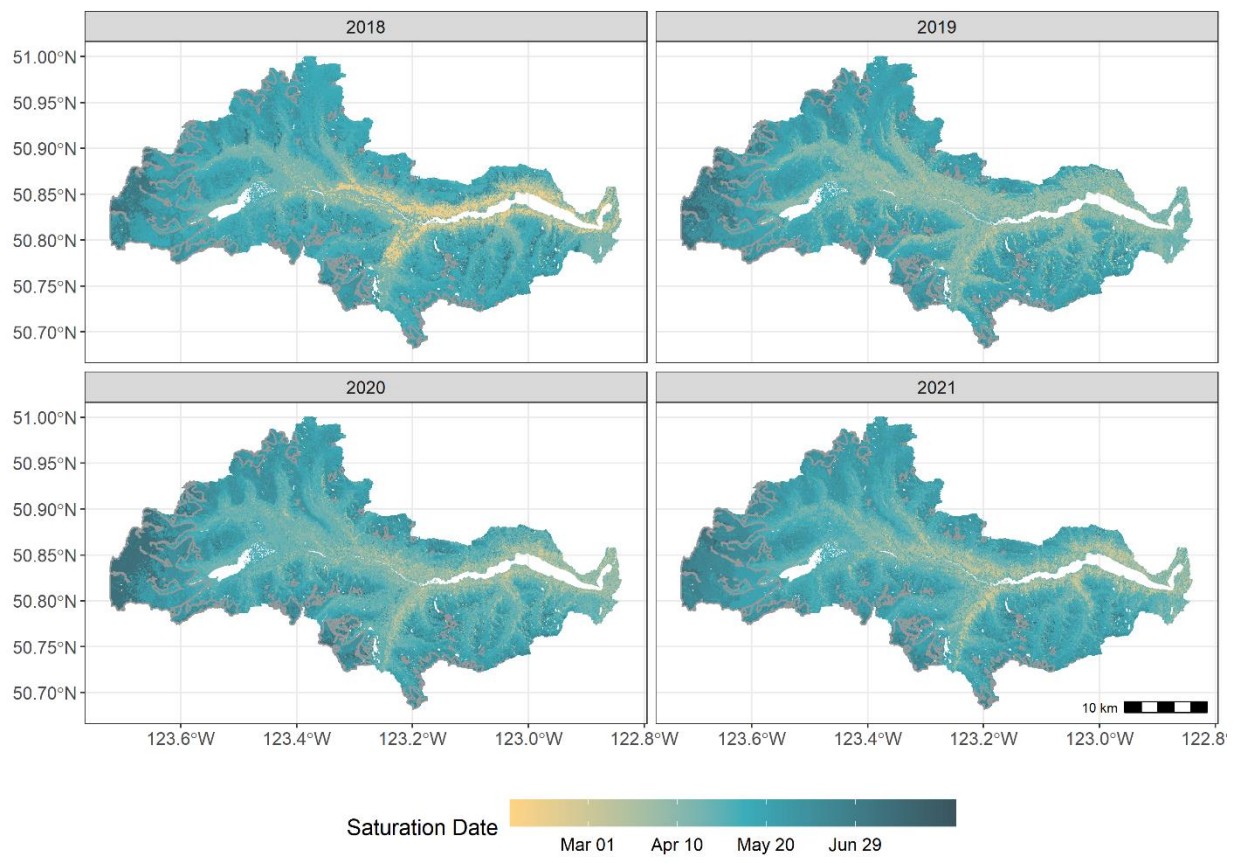

**Figure 6.** Estimates of snowpack saturation (snowmelt onset) from Sentinel-1 SAR in the Lajoie Basin. Snowmelt onset estimates are inferred from minima in SAR time series.

Generally, SAR estimates of melt onset indicate that snowmelt initiates in early March at low elevations in the Lajoie, with later snowmelt onset dates observed at higher elevations. The highest elevation pixels are estimated to initiate melt in mid-July. When examined by elevation, 2018 showed the earliest average melt onset dates when compared to other study years, with low elevations estimated to initiate melt as early as March 15[th]. At low elevations, pixels were estimated to initiate melt the latest in 2019 when compared to other study yeas, with estimates averaging as late as April 7[th]. The largest range in mean


melt onset estimates was produced in 2018, with 97 days between the earliest and latest elevation-binned averages. The
smallest range was produced in 2019, with 83 days between the earliest and latest elevation-binned averages. At high
elevations, the latest average melt onset dates were estimated in 2020 with pixels initiating melt on July 5th. The earliest average
melt onset dates at high elevations were estimated in 2018 with the highest elevation pixels initiating melt on June 20th. Melt
onset estimates across elevations were influenced by aspect, slope, and land cover (Figure 7). From 2018 to 2021, south and
west facing slopes-initiated melt earlier than north and east facing slopes.  SAR melt onset estimates indicate that melt initiates
earlier in the Lajoie on steep slopes (40-49°) and initiates progressively later as slope decreases. Melt onset dates are similar
between land cover types, with the exception of seasonal snow on glaciers which initiated later than other classes in all study
years.

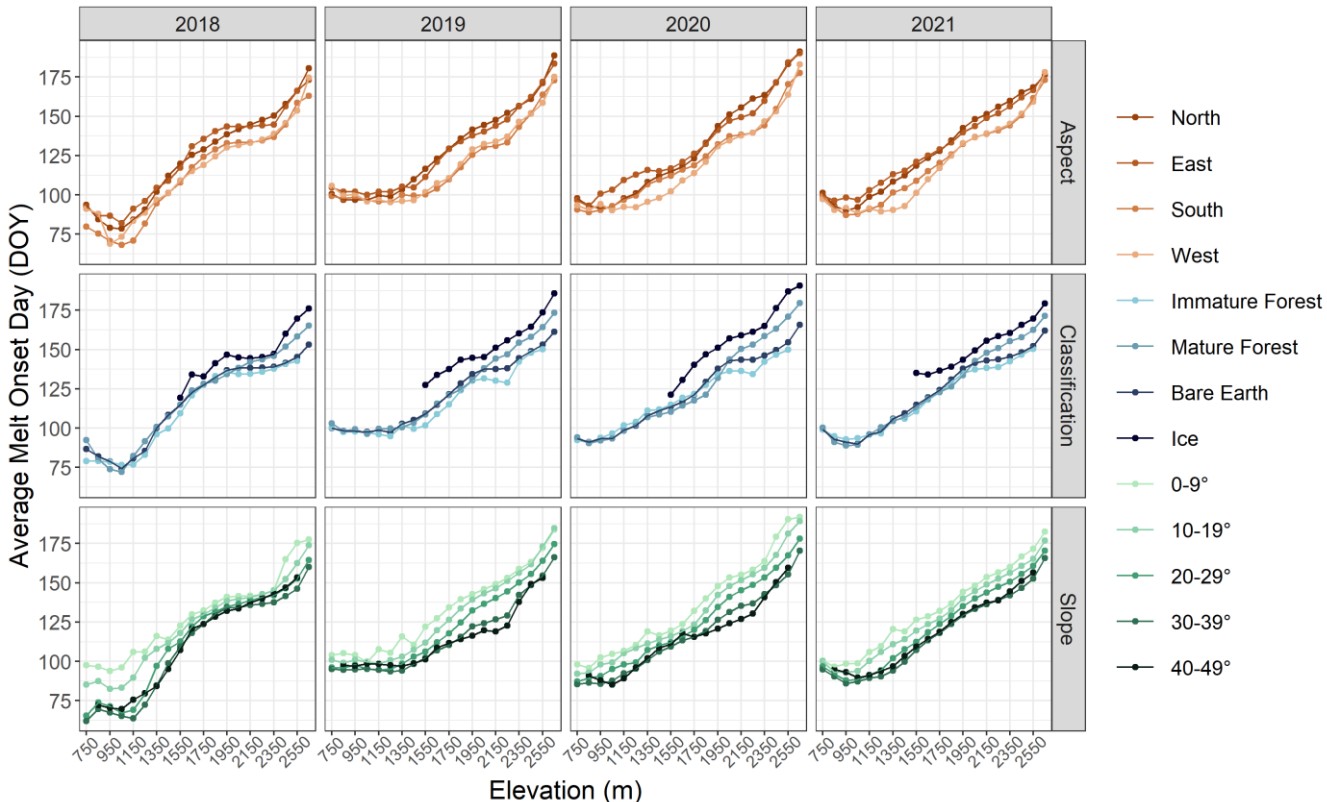

**Figure 7.** Average estimates of snowmelt onset by elevation, aspect, classification, and slope. Estimates of melt onset are inferred from
minima in Sentinel-1 SAR time series.

**5.4 Optical Estimates of Snow Disappearance**

Snow free dates estimated from optical and multispectral imagery (Figure 8) show a similar elevation dependence as melt
onset. However, we observe a greater variability in the snow free dates in forested regions and at lower elevations.


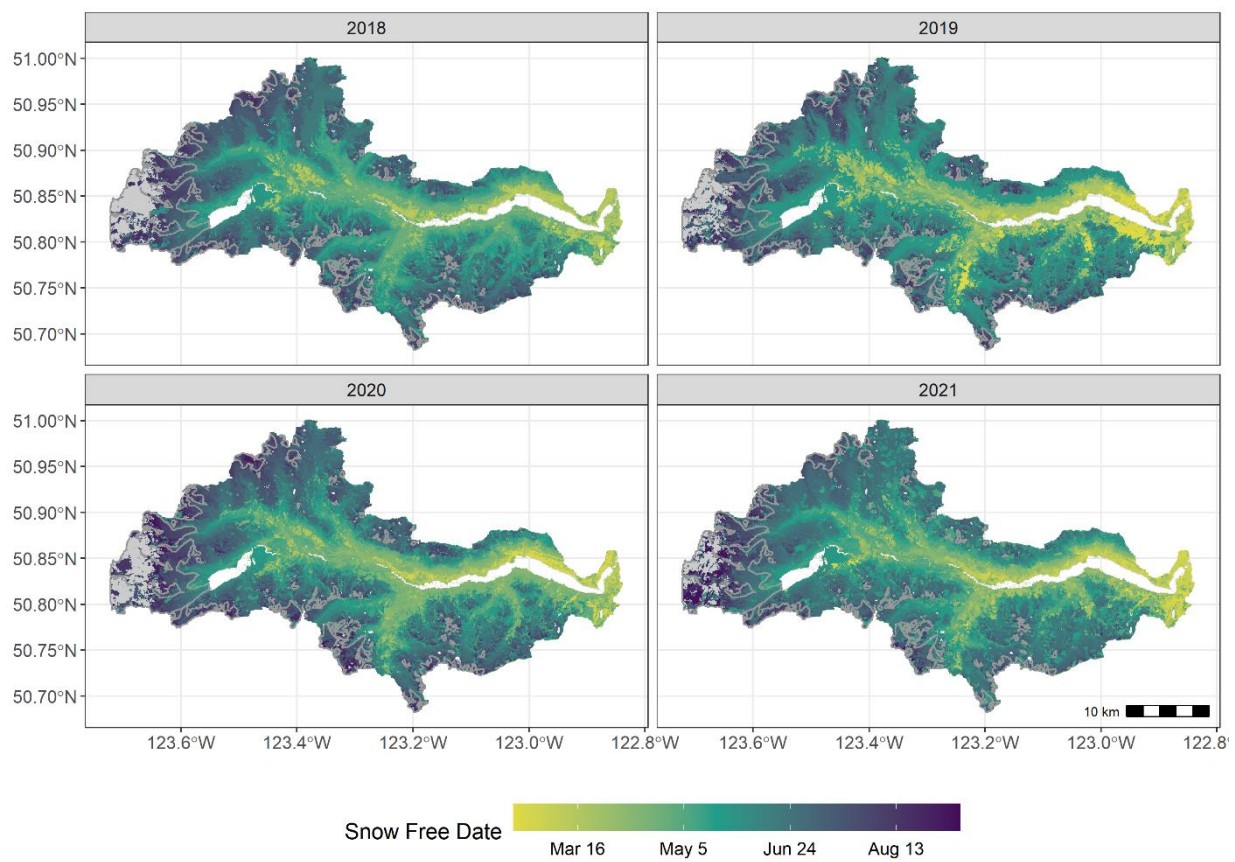

**Figure 8.** Estimates of snow free dates in the Lajoie Basin from Landsat-8 and Sentinel-2. Gray shading represents perennial snow, and the gray outlines delineate glacierized areas.

The detection of snow free dates from Landsat-8 and Sentinel-2 images provided consistent estimates by aspect, elevation,
land cover type and slope (Supplementary Materials). The impact of aspect is most pronounced at upper elevations, with northern and eastern slopes melting later than southern and western slopes. Similarly to SAR estimates of melt onset, snow disappearance estimates were the most variable on steep slopes (>30°). When compared to the other land classes, snow cover on glaciers disappears much later in the ablation season. Trends in snow free dates are consistent between study years.

**5.5 Snowmelt Duration Products**

With SAR-derived estimates of melt onset and optical/multispectral estimates of snow disappearance, we map snowmelt duration in the Lajoie Basin (Figure 9). SAR snowmelt duration estimates formed consistent patterns based on elevation



(Supplementary Materials). Snowmelt duration estimates were the longest between elevations of 2200 and 2400 m asl in the Lajoie from 2018 to 2021. Between 2200 and 2400 m asl the snowpack produced runoff for 46 days on average, whereas the basin-wide average was 30 days. At elevations above 2400 m asl duration estimates shorten to an average of 38 days. At low

elevations (<1200 m asl) estimated snowmelt durations are the shortest, lasting 14 days on average. Between study years, elevation-averaged snowmelt duration estimates were the shortest in 2021 at 27 days, on average, and the longest in 2018 at 32 days, on average.

Snowmelt durations are more sensitive to land cover, slope, and aspect when compared to melt onset dates, especially at elevations of greater than 1400 m asl. Melt durations were longer on northern and eastern facing slopes compared to southern

and western slopes. Increasing slope decreases melt durations in the Lajoie, with the longest durations observed in relatively flat areas (0-9°) above 2000 m asl. Land cover strongly influences melt durations. On average, from 2018 to 2021 snowmelt duration estimates were the longest over glacierized terrain, followed by mature forest. Snowmelt duration estimates were the shortest over disturbed or immature forest, followed by bare earth.



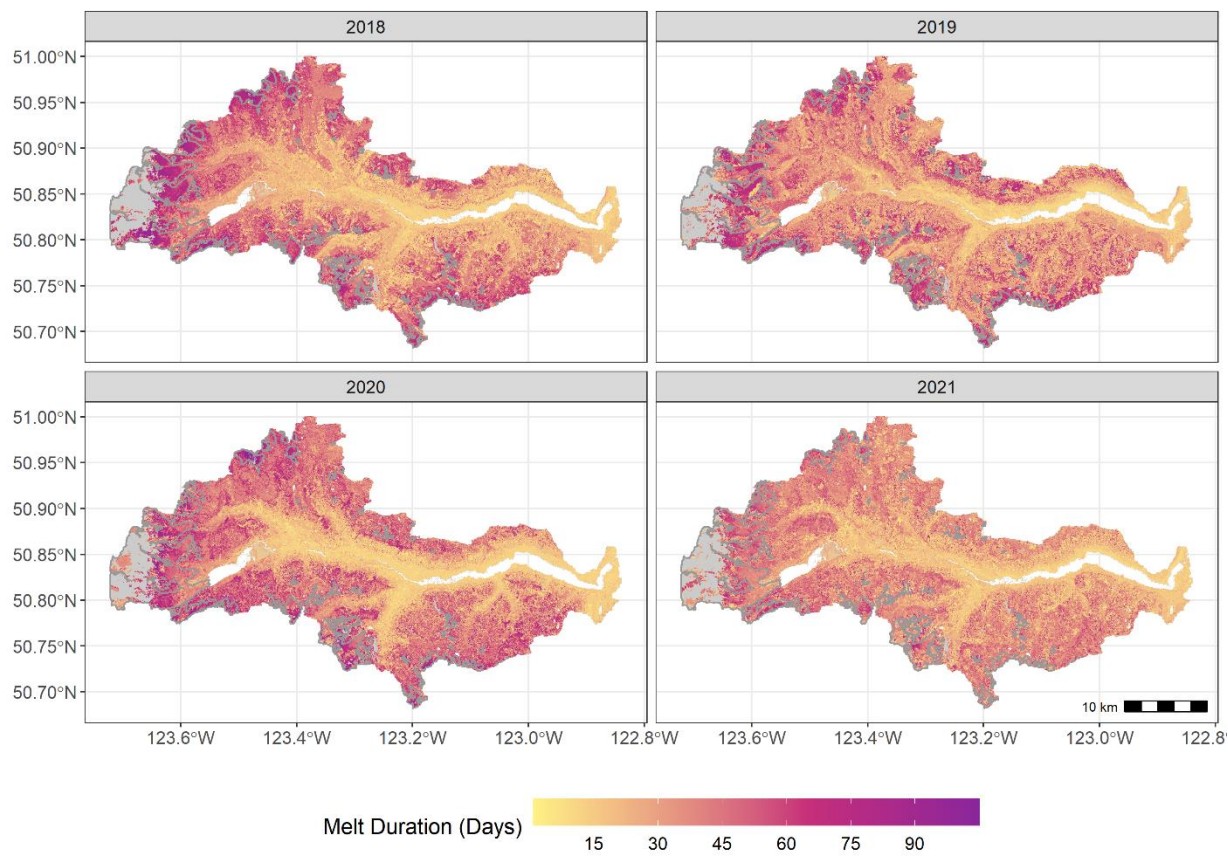

**Figure 9.** Snowmelt runoff duration in the Lajoie Basin from data fusion melt products. Durations are approximated by differencing radar estimates of snowmelt onset and optical/multispectral estimates of snow free dates. Gray shading represents perennial snow, and the gray outlines delineate glacierized areas.

Snowmelt duration estimates from data fusion did not always reflect melt durations from SWE records at automated weather stations (Table 5, Table 6). From data fusion, melt durations ranged from 19 to 38 days at Downton Lake Upper and from 14 to 36 days at Green Mountain. Higher inaccuracies are seen in the optical estimates of snow free dates when compared to SAR estimates of snow melt onset.




**Table 5.** Comparison of data fusion and SWE estimates of snowmelt end, onset and duration at Downton Lake Upper. The difference in days is provided between telemetry based (TLM) estimates of melt, and SAR, optical (OPT) and data fusion estimates of melt.

|  | 2018 | 2019 | 2020 | 2021 |
|---|---|---|---|---|
| **Onset (TLM)** | May 5 | May 8 | May 4 | May 10 |
| **Onset (SAR)** | May 12 | May 7 | May 13 | May 19 |
| **Difference (Days)** | 7 | 1 | 9 | 10 |
| **Melt End (TLM)** | June 24 | June 17 | June 24 | June 26 |
| **Melt End (OPT)** | June 19 | June 14 | June 16 | June 8 |
| **Difference (days)** | 5 | 3 | 8 | 18 |
| **Duration (TLM)** | 50 | 40 | 51 | 47 |
| **Duration (Fusion)** | 38 | 38 | 34 | 19 |
| **Difference (days)** | 12 | 2 | 17 | 28 |

**Table 6.** Comparison of data fusion and SWE estimates of snowmelt end, onset and duration at Green Mountain. The difference in days is provided between telemetry based (TLM) estimates of melt, and SAR, optical (OPT) and data fusion estimates of melt.

|  | 2018 | 2019 | 2020 | 2021 |
|---|---|---|---|---|
| **Onset (TLM)** | April 29 | April 30 | May 7 | May 13 |
| **Onset (SAR)** | May 12 | May 19 | May 4 | May 20 |
| **Difference (Days)** | 13 | 18 | 3 | 7 |
| **Melt End (TLM)** | June 3 | June 3 | June 9 | June 9 |
| **Melt End (OPT)** | June 17 | March 29 | May 18 | June 1 |
| **Difference (days)** | 14 | 66 | 22 | 7 |
| **Duration (TLM)** | 35 | 33 | 33 | 27 |
| **Duration (Fusion)** | 36 | 32 | 14 | 27 |
| **Difference (days)** | 1 | 1 | 19 | 0 |





# 6 Discussion

## 6.1 Role of Slope, Cover and Satellite Polarization in Snowmelt Onset Estimates

While VV polarized time series produced more accurate estimates of melt compared to VH polarized time series at automated weather stations, increased noise in VV time series is observed in the Lajoie (Figure 5). The amplified noise in the VV time series can be attributed to the steep slopes and forest cover present in the Lajoie. Vegetation covers a significant proportion (~45%) of the total area of the Lajoie Basin. Manickam and Barros (2021) report that a common thresholding approach for mapping wet snow using co-polarized SAR images (Nagler and Rott, 2000) failed in Colorado, USA, due to the presence of

conifer forest. Backscatter coefficients from cross-polarization were more sensitive to snow cover below vegetation compared to those from co-polarization and enabled wet snow mapping below the treeline (Manickam and Barros, 2020). Furthermore, the high relief in our study area may amplify the observed noise in co-polarized images, as steep slopes reduce the ability of VV polarization to distinguish between wet snow and snow free surfaces (Nagler et al., 2016). The combination of steep slopes and forested regions present in the Lajoie increase noise in VV time series; however, VV polarized SAR images are shown to

be more sensitive to wet snow compared to VH polarized images (Nagler et al., 2016). The increased sensitivity to wet snow in VV polarized images is reflected in the Lajoie through the greater accuracy of VV time series for approximating snowmelt onset at Downton Lake Upper and Green Mountain.

  The 'U' shape of SAR time series is more pronounced in open areas when compared to mature forest (Figure 5). As a result, estimates of melt onset and duration were less reliable in forested areas. The decreased sensitivity in forested areas can be

attributed to the scattering of SAR signals by the forest canopy. Forests effectively scatter radar energy depending on their structure, composition, and stem density (Bernier, 1987). Conifers, in particular, strongly scatter C-band SAR signals (Bernier, 1987). Spruce and fir trees (conifers) dominate forests in the Lajoie, leading to increased noise in melt onset maps from Sentinel-1 data. The Locally Weighted Least Squares Regression temporal smoother helped to reduce the variability of melt onset estimates in forested regions; however, ground truthing in these regions is required to verify the validity of this shift.

In open areas, the 'U' shape of SAR time series is more pronounced on flat slopes when compared to steep slopes (Figure 5). Manickam and Barros (2020) presented similar findings in the Swiss Alps, where between April and May backscattering coefficients were less sensitive to wet snow as slope increased. In forested regions, however, slope has a less uniform effect on SAR signals. The most distinct 'U' shaped signals are created from SAR time series collected on steep (>30°) slopes and the least distinct are collected on moderate (10 - 29°) slopes. Scatter from forests is impacted by terrain, which induces changes

in Trunk-Ground and Crown-Ground scattering mechanisms (Park et al., 2012). Vegetative scattering mechanisms can create more negative values based on ground surface tilt (Park et al., 2012). In the Lajoie, the increased visibility of SAR minima on steep slopes in forested regions may be related to terrain induced changes to vegetative scattering mechanisms.





## 6.2 Interannual Variability in Data Fusion Melt Products

Data fusion estimates of melt onset and duration were impacted by interannual climate fluctuations. At Downtown Lake Upper

from 2018 to 2021 SAR estimates of melt onset from VV timeseries were within 0-4 days of telemetry estimates. For VH polarized time series, melt onset estimates were within 0-4 days of telemetry estimates in 2018 and 2021 only, with minima occurring 10 days and 9 days early in 2019 and 2020, respectively. At Green Mountain, VH estimates of melt onset were also early in 2020, with minima occurring 13 days before telemetry estimates. VV and VH estimates were late in 2019 at Green Mountain, with both minima occurring 11 days after telemetry estimates. In 2020 early estimates of melt onset at both stations

may be the product of large temperature fluctuations in early April. Melt-refreeze cycles alter snowpack LWC and grain size (Yamaguchi et al. 2010), and thus will impact SAR signals (Liu et al., 2006). The late and early melt onset estimates in 2019 are also a likely result of temperature fluctuations. In 2019, day and night time temperatures at the Lajoie exceeded 0 °C from March 17th to 22nd, increasing snowpack LWC and thus the possibility for early melt onset estimates. However, after the initial warming period in 2019 temperatures decreased, with average night time temperatures of below 0 °C for the entire month of

April. Snowpack refreezing can decrease seasonal differences in SAR backscatter values (Floricioiu & Rott, 2001), and, as a result, minima in SAR timeseries may become less pronounced during refreezing periods. Between study years, interannual variations in climate, and therefore snowpack metamorphosis, influenced the accuracy of SAR snowmelt retrieval in the Lajoie and can partially explain the variability in snowmelt onset at automated weather stations.

## 6.3 Validation of Data Fusion Melt Products

Melt onset and duration estimates displayed varying agreement with continuous SWE records at Downton Lake Upper and Green Mountain. While all VV SAR minima from the extracted time series were within 5 days of melt onset at Downton Lake Upper, they occurred up to 11 days apart at Green Mountain. Green Mountain is located on a northeast facing slope, and receives morning sunlight. For this analysis only descending images were used, which are captured around 0700 PST in the Lajoie. Descending images in the Lajoie may be more susceptible to false positives from morning wetting, on east facing

slopes, during the ripening stage. Morning wetting of the snowpack may attribute to the greater variability seen in Green Mountain time series when compared to Downton Lake Upper. At both locations, time series estimates of melt onset were improved in VV polarization when compared to VH polarization. Final melt onset maps, however, were more accurate from VH polarized melt onset estimates when compared to VV, with 75% of VH estimates at Green Mountain occurring within one day of telemetry estimates (Supplementary Material). Co-polarized SAR signals have shown greater sensitivity to wet snow

when compared to cross-polarized images (Nagler et al., 2016); however, cross-polarized images are preferable when mapping wet snow below the treeline (Nagler et al., 2016; Manickam and Barros, 2020). Green Mountain is more heavily vegetated compared to Downton Lake Upper, which may result in the increased accuracy of VH estimates at this location. As the accuracy of VH observations at Green Mountain in final melt onset maps was not mirrored in time series estimates, increased





physical observations of the snowpack are required to further quantify the impact of polarization when using SAR minima as

proxies for snowmelt onset.

Inaccuracies in snowmelt durations at automated weather stations are largely attributed to errors in snow free estimates from optical and multispectral imagery. Snowpack monitoring with optical remote sensing data is challenging in the Lajoie, as imagery is frequently obscured by cloud cover. Cloud cover is persistent at high elevations during the ablation season in the Lajoie Basin. Missing data from cloud cover reduces the temporal resolution at which imagery is available, decreasing the

accuracy of snow disappearance estimates. The inclusion of optical data with a more frequent revisit interval could improve this analysis.

Despite errors in snow free estimates, the observed trends in snowmelt durations agree with trends in snow depths observed in the study area. The snowpack is generally deepest at mid to high elevations (2200 m asl to 2400 m asl) in the Lajoie Basin, as at the highest elevations there is heavy redistribution of snow by wind and avalanching.

SAR retrievals of snowmelt onset and duration rely on the high resolution and revisit frequency of Sentinel-1. On December 23rd, 2021, Sentinel-1B malfunctioned and has not been communicating data since (European Space Agency, 2022). The loss of Sentinel-1B reduces the number of observations available for snowpack monitoring, and may not be sufficient to capture melt onset until the launch of Sentinel-1C in 2023. Alternative C-Band SAR datasets can be explored for snowmelt monitoring, such as the RADARSAT Constellation Mission.

**7 Conclusions**

We present a low-cost, adaptable method to estimate snowmelt onset and duration through the fusion of Sentinel-1, Sentinel-2, and Landsat-8 imagery. Estimates of snowmelt onset from Sentinel-1 SAR were in agreement with continuous SWE observations from automated weather stations in the Lajoie basin.  On the watershed scale, estimates of snowmelt onset in the Lajoie reflect changes in elevation and topography, and are sensitive to land cover and slope. While steep slopes introduce

error into SAR snowmelt onset estimates in open areas, they can improve minima detection under vegetative cover. To map snowmelt on the watershed scale VV polarized images are recommended; however, VH polarized images may produce more accurate results in forested areas. Although snowmelt durations agreed with snow depth records in the Lajoie, they were inaccurate at automated weather stations due to cloud cover in Sentinel-2 and Landsat-8 imagery.

To reduce errors in snowmelt durations in future studies, optical/multispectral imagery with a higher cadence is recommended.

Further, a dense network of field observations is required to validate estimates of snowmelt onset and duration, particularly under varied land cover types. Future research will explore the impact of forest cover on snowmelt estimates in greater depth and will provide additional validation for snowmelt estimates in alpine regions.

Snowmelt dominates hydraulic regimes in Western North America and thus requires frequent monitoring. Our study supports findings from Marin et al. (2020) that Sentinel-1 SAR can be used to characterize snowmelt in these regions We demonstrate

Sentinel-1 SAR observations provide high resolution estimations of snowmelt onset and can be used to characterize snowmelt


in ungauged basins. These findings are of increased importance as current snowmelt regimes are threatened by warming global temperatures, which alter the extent and the duration of snow cover (Mote et al., 2005). As changes in snowmelt timing impact freshwater availability and natural hazard risk, increased observations of the snowpack, such as those provided by radar satellites, are required moving forward.

## Code Availability

The code used to download Sentinel-1 SAR images for snowmelt analysis is freely available at: https://code.earthengine.google.com/59db19e48330480186fa9f02ff3b4efb. Code for estimating snow disappearance from Sentinel-2 and Landsat-8 images is available at: https://code.earthengine.google.com/b15169de078690dc39e60fe5f398da1f.

## Data Availability

Sentinel-1, Sentinel-2, and Landsat-8 image collections are available via the Google Earth Engine Data Catalogue: https://developers.google.com/earth-engine/datasets/catalog.

## Author Contribution

All authors contributed to research design and result interpretation. Analysis was performed by SD, with regular guidance from JS and AC. SD wrote the paper based on input and feedback from JS, AC, and BM. LiDAR data was provided by BM.

## Competing Interests

The authors declare that they have no conflict of interest.

## Acknowledgements

We gratefully acknowledge funding provided for this research by Mitacs, BC Hydro, and the Government of British Columbia. Automated weather stations in the Lajoie are maintained and operated by BC Hydro, and optical and SAR data are distributed by the European Space Agency and the USGS. LiDAR estimates of snow depth were processed by S. Tagle.

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
