# Peer review of "Snowmelt Characterization from Optical and Synthetic Aperture Radar Observations in the Lajoie Basin, British Columbia"

_The Cryosphere, 2022_

## Author Comment (AC1)

Referee #1 Response Letter – Manuscript tc-2022-89

We thank Dr. Marin for their constructive and thorough feedback on our manuscript. The comments were very helpful, and have led to improvements in the paper. Please see our following responses **in bold** and our proposed alterations.

Pag 7 line 173, It would be interesting to know how much the percentage of infilled areas is.

**The percentage of infilled areas was available in the Supplementary Material. We now moved it into the main manuscript for easier reference.**

Pag 7 line 173, It is not clear if the signature always developed also in the cases where the dates were outside of two and a half standard deviation.

**The SAR snowmelt signature varies within the infilled areas (i.e., those outside two and a half standard deviation). Backscatter does not decrease during ablation or increase after snowmelt has occurred in many infilled pixels. This reduced fidelity becomes increasingly pronounced as slope and elevation increases. Backscatter values within some infilled pixels drop during the ablation period and increase after melt has occurred (i.e., have a snowmelt signature), especially those on low slopes in open or glacierized areas, however. For universal application over the Lajoie, we used standard deviation to remove variable SAR onset estimates stemming from noise or insensitivities in backscatter values. Future research will develop a time series analyses approach to reduce error in melt onset estimates.**

Pag 8 line 198, The identification of the end of the season using sparsely acquired high resolution data is a very challenging task. Some methods have been presented in the literature that try to address this problem using HR multi-source (optical-optical and optical-SAR) data and it is worth to mentioning them

**To improve the detection of snow disappearance, we used Landsat 8 and Sentinel-2 images (optical-optical), however additional preprocessing, or more complex snow detection algorithms, may improve results. Due to the large volumes of data we processed, additional optical image processing, or optical-radar fusion, was beyond the scope of the study. Line 405 has been amended and the following passage has been added:**

**"The inclusion of optical data with a more frequent revisit interval or the use of complex snow detection algorithms could improve this analysis. Support vector machine algorithms have yielded promising results for multi-source snow detection with SAR-optical data in mountainous environments (He et al. 2015, Lui et al. 2020). Further, downscaling and gap filling algorithms can improve snow cover detection using multi-source optical data (Premier et al. 2021, Revuelto et al. 2021). Revuelto et al. (2021) produced 20 m snow cover products by downscaling observations from the high-frequency, low-resolution Moderate Resolution Imaging Spectroradiometer with Sentinel-2 observations. In future studies, downscaling or machine learning methods can be adopted to reduce error in snow disappearance estimates."**

Table 2, It is not clear, at least to me in the present form, what the image frequency is. For example 1-7 days refer to the day in which there is an acquisition?

**The image frequency here refers to the number of days between image acquisitions. For example, in 2021 from the descending orbit there was a minimum gap between acquisitions of one day, and a maximum gap of 12 days. The chart has been changed to 'Acquisition Frequency'.**

Table 3, It would be interesting to know the percentage of image for which the cloud cover is less than a given low threshold e.g., 30%.

**For optical images, we used approximately half of all available Sentinel-2 and Landsat-8 images. In 2018, 217 images were available with 53% falling below the 60% cloud cover threshold. In 2019, 2020, and 2021, 51%, 56%, and 55% of images were used, respectively. We added the following sentence (line 184):**

**"Images were further filtered to remove scenes with more than 60% cloud cover, resulting in the use of 53%, 51%, 56% and 55% of all available images from 2018-2021, respectively."**

Figure 2, It is not clear why the shaded blue representing the melt period is not stopping at SWE = 0 for some years. I think this is the rule to be applied once the onset is identified.

**In Figure 2 there was an error in the graph, the shaded blue area for the Green Mountain station was displayed on the graph for the Downton Lake Upper station. The shaded area has now been corrected and updated in the manuscript. We chose to use the final breakpoint in the piecewise linear regression to mark the end of the melt period as SWE does not reach zero. After the end of snowmelt, the observed SWE often ranges from 5 mm to 15 mm due to error in the snow pillows. The final breakpoint was used instead as it can be applied universally between the stations and study years.**

Interestingly the onset for the runoff is derived in simplified snow model by considering the average temperature (and the radiation) i.e., degree day model. If air temperatures are available for the Lajoie basin, it would be interesting to discuss the difference between Sentinel-1 in identifying the runoff onset (temperature can be spatialized at high resolution and thresholded accordingly).

**Thank you for this suggestion – using a simple degree day model (Eq. 1), we now compare SAR estimates of melt onset with modelled estimates based on extrapolated air temperatures from the Downton Lake Upper snow pillow.**

$$M = C_m(T_a - T_b) \hspace{3cm} \text{Eq. 1}$$

where M is snowmelt in mm $d^{-1}$, $C_m$ is a degree-day coefficient in mm $day^{-1}$ $°C^{-1}$,

$T_a$ is the mean daily air temperature in °C, and $T_b$ is the threshold temperature in °C. For this analysis, we used a degree-day coefficient of 3.2 mm day$^{-1}$ °C$^{-1}$, based on reported values from North American and maritime snowpacks (DeWalle et al. 2002, Hock 2003). We selected a threshold temperature of 0°C, which is commonly used in snowpack studies (Zhou et al. 2021). We extrapolated air temperatures from the Downton Lake Upper station using a lapse rate of 6.5°C km$^{-1}$. We identified melt onset per pixel as the first date in the ablation season where accumulated melt is only positive (i.e., the date after the last negative or zero melt value).

On average, SAR predicts earlier melt onset compared to results from the degree day model (Figure 1). These discrepancies may stem from the melt model and its assumptions, the inability of temperature-based models to account for aspect, slope, and terrain cover, or errors in the SAR-derived melt onset estimates.

[Figure]

Figure 1. Comparisons of average melt onset dates from SAR and modelled estimates. Modelled estimates are obtained from a simple degree day model using temperature data from the Downtown Lake Upper snow pillow.

At low elevations the Lajoie is forested. In warm maritime environments, forest cover can enhance snowmelt at low elevations due to increased longwave radiation and warmer air temperatures (Lundquist et al. 2013). Forest cover can also prevent longwave radiative cooling at night, leading to winter snowpacks that are closer to 0°C. In these low-elevation forests, the earlier onset estimates from SAR, when compared to the degree-day model, may capture the effects of forests on melt onset. These effects would not be accounted for in a temperature-based model. At high elevations in the Lajoie, snow melt may be accelerated due to the greater prevalence of steep slopes. As steep slopes tend to accumulate less snow, they may melt earlier compared to surrounding areas which a temperature-based model can not account for. Further, surface melt can occur when temperatures are below 0°C, leading to the underestimation of melt when using 0°C as the threshold temperature.

Additional analyses are needed to determine how model calibration, variation in lapse rates, and topography impact results, and is beyond the scope of the project. Future work will compare SAR and modelled estimates of snowmelt onset in greater depth.

The sampling time provided by Sentinel-1 seems to be not adequate, in Shannon sense, to properly sampling the melting which has probably a temporal resolution less than one day. That means that the error could be potentially of several days. How is this uncertainty propagating in the case of snow melt duration analysis when different years are compared? What is the ideal revisit time needed for this kind of analysis?

Snowmelt durations are impacted by both errors in SAR onset estimates and optical snow disappearance estimates. Between study years SAR acquisitions are available every five to twelve days, averaging 6 days in each year. Optical observations are more variable between study years, and are summarized in Table 1.

Table 1. Revisit intervals for optical imagery from Sentinel 2 and Landsat 8 for the Lajoie Basin.

| Year | Minimum Revisit Interval | Maximum Revisit Interval | Average Revisit Interval |
|------|--------------------------|--------------------------|--------------------------|
| 2018 | 1 day | 17 days | 4 days |
| 2019 | 1 day | 15 days | 4 days |
| 2020 | 1 day | 18 days | 3 days |
| 2021 | 1 day | 8 days | 3 days |

We calculated errors in snowmelt durations ($\sigma_D$) using Equation 2.

$$\sigma_D = \sqrt{\sigma_{ONS}^2 + \sigma_{END}^2} \qquad\qquad \text{Eq. 2}$$

where error in the onset ($\sigma_{ONS}$) is six days, and error in snow disappearance ($\sigma_{END}$) is the average revisit interval from the annual Landsat 8 and Sentinel 2 image collections. Rounding to the nearest day, this yields an average error in the duration estimates of ±7 days for all study years. Ideally, a revisit time for SAR observations of three days would reduce this error to ±4 days. Daily SAR observations could reduce error in durations to ± 3 days; however, for reduced error, or operational applications, high-resolution optical imagery is needed at more frequent revisit intervals.

Table 3 (now Table 4 due to revisions) has been updated to include the revisit frequency of optical observations. Line 401 has been amended, and the following passage has been added.

"Inaccuracies in snowmelt durations are attributed to errors in SAR onset estimates and optical snow disappearance estimates. Between study years SAR acquisitions are available every five to twelve days, averaging six days in each year. Optical observations are variable between study years (Table 4); however, average revisit intervals are consistent at four days in 2018 and 2019

and three days in 2020 and 2021. We calculated errors in snowmelt durations ($\sigma_D$) using Equation 1.

$$\sigma_D = \sqrt{\sigma_{ONS}^2 + \sigma_{END}^2} \qquad\qquad \text{Eq. 1}$$

where error in the onset ($\sigma_{ONS}$) and disappearance dates ($\sigma_{END}$) are taken as the average revisit interval for each data type and study year. Rounding to the nearest day, this yields an average error in the duration estimates of ±7 days for all study years. Ideally, a revisit time for SAR observations of three days would reduce this error to ±4 days. Daily SAR observations could reduce error in durations to ± 3 days; however, for reduced error, or operational applications, high-resolution optical imagery is needed at more frequent revisit intervals."

References

DeWalle, D. R., Z. Henderson and A. Rango. 2002. Spatial and temporal variations in snowmelt degree-day factors computed from snotel. In Proceedings of the 70th Annual Meeting of the Western Snow Conference, Granby, CO, USA: 73–81.

He, G., P. Xiao, X. Feng, X. Zhang, Z. Wang and N. Chen. 2015. Extracting snow cover in Mountain areas based on SAR and optical data. IEEE Geoscience and Remote Sensing Letter, 12(5): 1136-1140.

Hock, R. 2003. Temperature index melt modelling in mountain areas. Journal of Hydrology, 282: 104-115.

Lui, Y., X. Chen, JS. Hao and L. Li. 2020. Snow cover estimation from MODIS and Sentinel-1 SAR data using machine learning algorithms in the western part of the Tianshan Mountains. Journal of Mountain Science, 17(4): 884-897.

Lundquist, J. D., S. E. Dickerson-Lange, J. A. Lutz and N. C. Cristea. 2013. Lower forest density enhances snow retention in regions with warmer winters: A global framework developed from plot-scale observations and modeling. Water Resources Research, 49: 6356-6370.

Premier, V., C. Marin, S. Steger, C. Notarnicola & L. Bruzzone. 2021. A novel approach based on a hierarchical multiresolution analysis of optical time series to reconstruct the daily high-resolution snow cover area. IEE Journal of Selected Topics in Applied Earth Observation and Remote Sensing, 14: 9223-9240.

Revuelto, J., E. Alonso-Gonzalez, S. Gascoin, G. Rodriguez-Lopez and J. I. Lopez-Moreno. 2021. Spatial downscaling of MODIS snow cover observations using Sentinel-2 snow products. Remote Sensing 13(22): 4513.

---

## Author Comment (AC2)

Referee #2 Response Letter – Manuscript tc-2022-89

We thank Dr. Bertoldi for their constructive and helpful feedback on our manuscript. Please see our responses and proposed alterations **in bold.**

Introduction. Please define in a more precise way the paper's aims and formulate clear research questions.

**To define our project's aim and research questions more precisely the Introduction has been modified starting on Line 78:**

**"We examine how the relation between SAR minima and snowmelt (Marin et al., 2020) is impacted by polarization, land cover, aspect, and hillslope in the Lajoie Basin. We further assess how estimates of snowmelt onset and duration can be verified with continuous records of SWE."**

Section 4.2 snow disappearance times. Since Landsat has a low overpass time and images are often cloud covered, please better quantify the errors and uncertainties in days for the snow disappearance time.

**Thank you for suggesting this. Reviewer One was also concerned about error propagation in snow disappearance estimates. Please see the last response in our reply to Reviewer One for further clarification:**

**Snowmelt durations are impacted by both errors in SAR onset estimates and optical snow disappearance estimates. Between study years SAR acquisitions are available every five to twelve days, averaging 6 days in each year. Optical observations are more variable between study years, and are summarized in Table 1.**

**Table 1. Revisit intervals for optical imagery from Sentinel 2 and Landsat 8 for the Lajoie Basin.**

| Year | Minimum Revisit Interval | Maximum Revisit Interval | Average Revisit Interval |
|---|---|---|---|
| 2018 | 1 day | 17 days | 4 days |
| 2019 | 1 day | 15 days | 4 days |
| 2020 | 1 day | 18 days | 3 days |
| 2021 | 1 day | 8 days | 3 days |

**We calculated errors in snowmelt durations ($\sigma_D$) using Equation 2.**

$$\sigma_D = \sqrt{\sigma_{ONS}^2 + \sigma_{END}^2} \qquad \text{Eq. 2}$$

**where error in the onset ($\sigma_{ONS}$) is six days, and error in snow disappearance ($\sigma_{END}$) is the average revisit interval from the annual Landsat 8 and Sentinel 2 image collections. Rounding to the nearest day, this yields an average error in the duration estimates of ±7 days for all study years.**

Ideally, a revisit time for SAR observations of three days would reduce this error to ±4 days. Daily SAR observations could reduce error in durations to ± 3 days; however, for reduced error, or operational applications, high-resolution optical imagery is needed at more frequent revisit intervals.

Table 3 (now Table 4 due to revisions) has been updated to include the revisit frequency of optical observations. Line 401 has been amended, and the following passage has been added.

"Inaccuracies in snowmelt durations are attributed to errors in SAR onset estimates and optical snow disappearance estimates. Between study years SAR acquisitions are available every five to twelve days, averaging six days in each year. Optical observations are variable between study years (Table 4); however, average revisit intervals are consistent at four days in 2018 and 2019 and three days in 2020 and 2021. We calculated errors in snowmelt durations ($\sigma_D$) using Equation 1.

$$\sigma_D = \sqrt{\sigma_{ONS}^2 + \sigma_{END}^2}$$   Eq. 1

where error in the onset ($\sigma_{ONS}$) and disappearance dates ($\sigma_{END}$) are taken as the average revisit interval for each data type and study year. Rounding to the nearest day, this yields an average error in the duration estimates of ±7 days for all study years. Ideally, a revisit time for SAR observations of three days would reduce this error to ±4 days. Daily SAR observations could reduce error in durations to ± 3 days; however, for reduced error, or operational applications, high-resolution optical imagery is needed at more frequent revisit intervals."

Section 5.2 Sensitivity - L233 - "the least accurate approximation in 2019" - please quantify in numbers, variance …

SAR time series estimates are now quantified using the difference in days between SAR melt onset estimates and telemetry records. Section 5.2 has been modified, starting on Line 233:

"At Downton Lake Upper, SAR time series minima occurred within 0-10 days of SWE melt onset estimates for both polarizations (Figure 3). Comparing across four melt seasons at Downton Lake Upper, the maximum difference (+4 days) between onset estimates from VV polarized time series and telemetry records was produced in 2019. The minimum difference was produced in 2018 when there was no difference between telemetry records of melt onset and SAR estimates of melt onset. The maximum difference (-10 days) between estimates from VH polarized time series and telemetry records was produced in 2020 at Downtown Lake Upper. The minimum difference was produced in 2018, when, similarly to VV time series, there was no difference between telemetry records of melt onset and SAR estimates of melt onset. At Green Mountain, SAR minima occurred within 1-13 days of SWE melt onset estimates (Figure 4). Between study years, the maximum difference (+11 days) between onset estimates from VV polarized timeseries and telemetry records was produced in 2019. The minimum difference was produced in 2019 (-1 day). The maximum difference (-13 days) between onset estimates from VH polarized time series and telemetry records was produced in 2020 at Green Mountain. The minimum

difference was produced in 2021, when there was no difference between telemetry records of melt onset and SAR estimates of melt onset."

L255 - besides slope, does the accuracy of results change also with the aspect?

The SAR snowmelt signature, or the decrease of backscatter during ablation and subsequent increase after snowmelt has occurred, varies in amplitude between aspects (Figure 1). While the snowmelt signature is discernable in all aspects, it is the most pronounced on eastern and northern slopes when compared to southern and western. This phenomenon may relate to the distribution of slopes within the Lajoie. We observe the largest amplitude of the snowmelt signal in open, low slopes (Figure 4 in the manuscript). Low gradients (i.e., < 20°) are the most prevalent on North and East facing slopes in the Lajoie and may contribute to the larger amplitude observed in these aspects. Satellite look angle may also influence the strength of the SAR snowmelt signature between aspects. Overall, SAR onset estimates are more consistent by aspect when compared to slope (Figure 7 in the manuscript) and were of lesser focus in the analysis.

[Figure]

**Figure 1. SAR backscatter time series in the Lajoie Basin from pixels located between 1600 and 1800 m from VV (top) and VH (bottom) polarized images. Observations under mature forest cover are displayed on the right, whereas observations in open areas are displayed on the left. Average backscatter for each cover type is shown by the shaded lines, with each line representing a different aspect. Observations are from 2021.**

Paragraph at line 315 - Interesting! Maybe an additional Figure can support this!

A figure supporting this was formerly in the Supplement (Figure S9). It has been moved into the main manuscript for easier reference.

L324 - What do you mean by Data Fusion? Please explain better!

We use data fusion to describe the process of combining remotely sensed datasets to estimate snow disappearance and snowmelt duration. To improve clarity, the term data fusion has been replaced with "multi-source" or "optical-radar" throughout the manuscript.

---

## Author Response (AR2)

**Manuscript tc-2022-89 – Response Document**

Comments to Referee Three

We thank Referee Three for their feedback on our manuscript. The comments were helpful and have led to improvements in the paper. Please see our following responses **in bold** and our proposed alterations.

Throughout the manuscript, the author should update their date formatting to that stated by The Cryosphere, "Date and time: 25 July 2007 (dd month yyyy)"

**Upon consultation with the editor, we have decided to leave the dates formatted as is. We are concerned about plot space with the long date format, as well as redundancy with column and subplot headings.**

Figure 10 caption states, "Average estimates of snowmelt duration..." but the y-axis is "Average Melt Onset Date (DOY)". I'm unsure if this is the incorrect plot or just a typo; please check.

**The y-axis label was incorrect for Figure 10. It has been edited to read "Average Melt Duration (Days)" alongside Figure S10 in the Supplementary Materials. We appreciate that this mistake was caught, as a similar issue was found with Figure S8. The y-axis of Figure S8 has now been adjusted to read "Average Day of Snow Disappearance (DOY)."**

L373: Change year from 2021 to 2020 as per citation.

**In the most recent upload of the manuscript, we were unable to find a citation from 2021 on L373. However, there was an error in the citation on L359. It has since been updated from 2021 to 2020.**

L446: Check this sentence's grammar - it probably needs and 'and'. "The inclusion of optical data with a more frequent revisit interval the use of complex snow detection algorithms could improve this analysis.

**In the most recent upload of the manuscript this sentence is located on L431. The sentence has been reworded for clarity: "The inclusion of optical data with a more frequent revisit interval, or the use of complex snow detection algorithms, could improve this analysis."**

Changes to the Manuscript

A summary of all relevant changes to the manuscript can be found here:

- **Line 359:** Citation is corrected to "2020."
- **Line 431:** Sentence amended to "The inclusion of optical data with a more frequent revisit interval, or the use of complex snow detection algorithms, could improve this analysis."
- **Figure 10:** The y-axis is changed to "Average Melt Duration (Days)."

- **Figure S8:** The y-axis is changed to "Average Day of Snow Disappearance (DOY)."
- **Figure S10:** The y-axis is changed to "Average Melt Duration (Days)."